# Prioritization of surgical, obstetric, trauma, and anesthesia care in India over seven decades: A systematic analysis of policy documents

Ritika Shetty[1,2‡], Siddhesh Zadey[1,3,4,5,6‡]*, Anushka Jindal[1,7], Himanshu Iyer[1,8], Sweta Dubey[1], Gnanaraj Jesudian[9,10], Emily R. Smith[3,4,5,11], Catherine A. Staton[3,4,5,11], Tamara N. Fitzgerald[3,11], Joao Ricardo Nickenig Vissoci[3,4,5,11]

**1** Association for Socially Applicable Research (ASAR), Pune, Maharashtra, India, **2** Terna Medical College and Hospital, Navi Mumbai, Maharashtra, India, **3** Department of Surgery, Duke University School of Medicine, Durham, North Carolina, United States of America, **4** Global Emergency Medicine Innovation and Implementation (GEMINI) Research Center, Duke University, Durham, North Carolina, United States of America, **5** Department of Emergency Medicine, Duke University School of Medicine, Durham, North Carolina, United States of America, **6** Dr. D.Y. Patil Medical College, Hospital, and Research Centre, Pune, Maharashtra, India, **7** King's College Hospital, Denmark Hill, London, United Kingdom, **8** Department of Surgery, Khurshitji Beharamji Bhabha Municipal General Hospital, Mumbai, Maharashtra, India, **9** Department of EIE, Karunya University, Coimbatore, Tamil Nadu, India, **10** Association of Rural Surgeons of India/International Federation of Rural Surgeons—Rural Surgery Research and Training Center, Shanthi Bhavan Medical Center, Jharkhand, India, **11** Duke Global Health Institute, Durham, North Carolina, United States of America

‡ RS and SZ are co-first authors on this work.
* sidzadey@asarforindia.org, siddhesh.zadey@duke.edu

**Data Availability Statement:** All data used and generated in this manuscript has been supplied in

## Abstract

Improving access to surgical care in India requires policy-level prioritization of surgical, obstetric, trauma, and anesthesia (SOTA) care. We quantified SOTA care prioritization in the last seven decades by analyzing India's national policy and programmatic documents. Forty documents of national importance over seven decades (1946–2017) were screened for a set of 52 surgical and 6 non-surgical keywords. The number of mentions per keyword was used as a proxy for surgical prioritization. For thematic analysis, surgical mentions were further classified into five domains: Infrastructure, Workforce, Service Delivery, Financing, and Information Management. The total number of mentions was 4681 for the surgical keywords and 2322 for non-surgical. The number of mentions per keyword was 90.02 for surgical keywords and 387 for non-surgical. The older committee reports showed relatively higher SOTA care prioritization compared to the years after 2010. Among the domains, Service Delivery (897) had the maximum number of mentions followed by Infrastructure (545), Workforce (516), Financing (98), and Information Management (40). National Health Policy 2017, the most recent high-level policy, grossly neglected SOTA care. SOTA care is inadequately prioritized in Indian national health policies, especially in the documents after 2010. Concerted efforts are necessary to improve the focus on financing and information management. Prioritization can be improved through a stand-alone national plan for SOTA care along with integration into existing policies.

the Supporting Information. See Supplementary
File 2.

**Funding:** The author(s) received no specific
funding for this work.

**Competing interests:** The authors have declared
that no competing interests exist.

## 1 Introduction

Globally, around 140 million people who need surgery are unable to access it [1]. Ensuring universal and equitable access to surgical, obstetric, trauma, and anesthesia (SOTA) care requires a scale-up of the surgical systems through adequate resource allocations. Resource allocation towards SOTA care is essential to prevent mortality due to surgical conditions in India. Such allocation would be possible if SOTA care issues receive adequate representation in health policy and planning. The Lancet Commission on Global Surgery (LCoGS) recommended formulating National Surgical, Obstetric, and Anesthesia Plans (NSOAPs) for low- and middle-income countries (LMICs) [2]. Further, ensuring the alignment of NSOAPs with existing health policies and providing pathways for integration of SOTA care in broader health planning is critical for successful implementation.

There is rising global awareness of SOTA care access. The 68th World Health Assembly passed a resolution on 'Strengthening Emergency and Essential Surgical Care and Anaesthesia' as a component of Universal Health Coverage (UHC) [3]. Resolution 68.15 which demands action from governments and policymakers around the world to address the gaps in SOTA care delivery and bolster public health infrastructure was signed by 194 countries including India. In 2015, through an analysis of intergovernmental health policy documents, LCoGS demonstrated that SOTA care gets limited attention compared to other global health issues such as HIV/AIDS, tuberculosis, and malaria [1]. The limited prioritization is particularly stark given that surgical conditions contribute to a greater mortality burden than these three conditions combined [4].

Since then, multiple analyses have attempted to understand SOTA care prioritization through document reviews. Analyses of national health policy and planning documents across 43 sub-Saharan African countries demonstrated limited focus on SOTA care and the need for better prioritization [5, 6]. Since then, several of these countries, including Zambia, Tanzania, Nigeria, and Rwanda have drafted and implemented their NSOAPs [7]. Another study found out the low importance given to pediatric surgery in national health policies, strategies, and plans across 128 countries has become an important instrument for global advocacy child surgery initiatives [8, 9]. These examples show that understanding the prioritization of SOTA care in policy discourse can provide valuable insights for future advocacy and planning. Such analysis is missing for India where 90% of 1.4 billion people lack SOTA care access and NSOAP development is currently lacking [10, 11]. This study aims to assess national-level SOTA care prioritization in relevant health policy, planning, and programmatic documents published from 1946 to 2017.

## 2 Materials and methods

### 2.1 Context of health policy and planning in India

National-level health policy-making and planning in India has been guided by recommendations from high-level government-appointed committees with programmatic guidelines driving implementation [12, 13]. Before the country's independence from colonial rule, the Bhore Committee (Health Survey and Development Committee) instituted under the British Raj, provided a comprehensive four-volume report in 1946 describing the country's needs and envisioning universal healthcare for India's future [14–16]. Subsequently, from 1962 to 1986, several committees dealing with specific health system issues were instituted including the Mudaliar Committee (health survey and planning), Mukerji Committee (delinking of malaria program from family planning activities), Jungalwalla Committee (unified cadre of health personnel), Kartar Singh Committee (multipurpose health workers for national programs),

Shrivastav Committee (medical education and manpower), and Bajaj Committee (health workforce) [17–23]. It is important to note that recommendations from these committees did not always make it to the agenda of those drafting policies [13].

Apart from committees, the Planning Commissions established by the Government of India played a major role in deciding the country's development agenda by formulating five-year plans. From 1951 to 2012, there have been twelve five-year plans with considerable focus on various aspects of healthcare. The 11[th] Planning Commission (2010) constituted a High Level Expert Group (HLEG) focusing on universal health coverage [24]. Another influence on the country's health policymaking has been its commitment to shared global goals [12]. For instance, after signing the Alma Ata Declaration of 1978, India drafted its first National Health Policy (NHP) in 1983 [25]. NHP provides a broad framework for the country's health priorities with mechanisms to achieve them. Since then, two iterations of NHP to address changing needs have come up in 2002 and 2017 coinciding with the global agenda for Millennium and Sustainable Development Goals (MDGs and SDGs) [26, 27].

The Ministry of Health and Family Welfare (MoHFW) launched the National Health Mission (NHM) in 2005, which later bifurcated into two sub-missions: National Rural Health Mission (NRHM) [28] and National Urban Health Mission (NUHM) [29]. NHM is aimed to provide access to affordable and quality healthcare through the public health system with a major focus on strengthening reproductive, maternal, newborn, child, and adolescent health. To benchmark the capacity of health facilities and the quality of care under NHM, the Indian Public Health Standard (IPHS) guidelines were drafted in 2007 and revised in 2012. Specific guidelines that state necessary and desirable parameters for health facilities at different levels of referral hierarchy are present for Sub-Centres (SCs) [30], Primary Health Centres (PHCs) [31], Community Health Centres (CHCs) [32], Sub-District Hospitals (SDHs) [33], and District Hospitals (DHs) [34]. Beyond these programs, India has a number of vertical initiatives targeted at specific health problems. Some main ones relevant to SOTA care include: National Blood Policy, National Program for Health Care of the Elderly, National Council for Clinical Establishments Report, National Oral Health Program, National Program for Palliative Care, National Program for Prevention and Management of Trauma and Burn Injuries, National Organ Transplant Program, National Program for Control of Blindness and Visual Impairment, and National Program for Prevention and Control of Cancer, Diabetes, Cardiovascular diseases, and Stroke [35–43]. All such policies and programs have been crucial for setting India's health agenda at the national level over the last 76 years.

## 2.2 Document selection

A systematic document review was conducted using a predefined list of keywords (see ahead) (**Fig 1**). National-level health-related policy, planning, and programmatic documents published during the period 1946 to 2017 were considered for screening. This duration was chosen as it covers the discourse from the first assessment committee to the most recent apex National Health Policy. Only health policy, planning, and programmatic documents that were relevant to various SOTA care themes present in the LCoGS, WHO Resolution, and the DCPN report were included for screening, while others such as the mental health policy or digital health mission were excluded. Health legislation and insurance-related documents were excluded. Documents in English that could be accessed through the internet, and that were searchable electronically were considered for inclusion. For selecting the relevant documents, an iterative search of various governmental platforms was conducted. Since a single repository for all such documents does not exist, we went through multiple government portals. The Ministry of Health and Family Welfare (MoHFW) website (https://main.mohfw.gov.in/) was assessed to

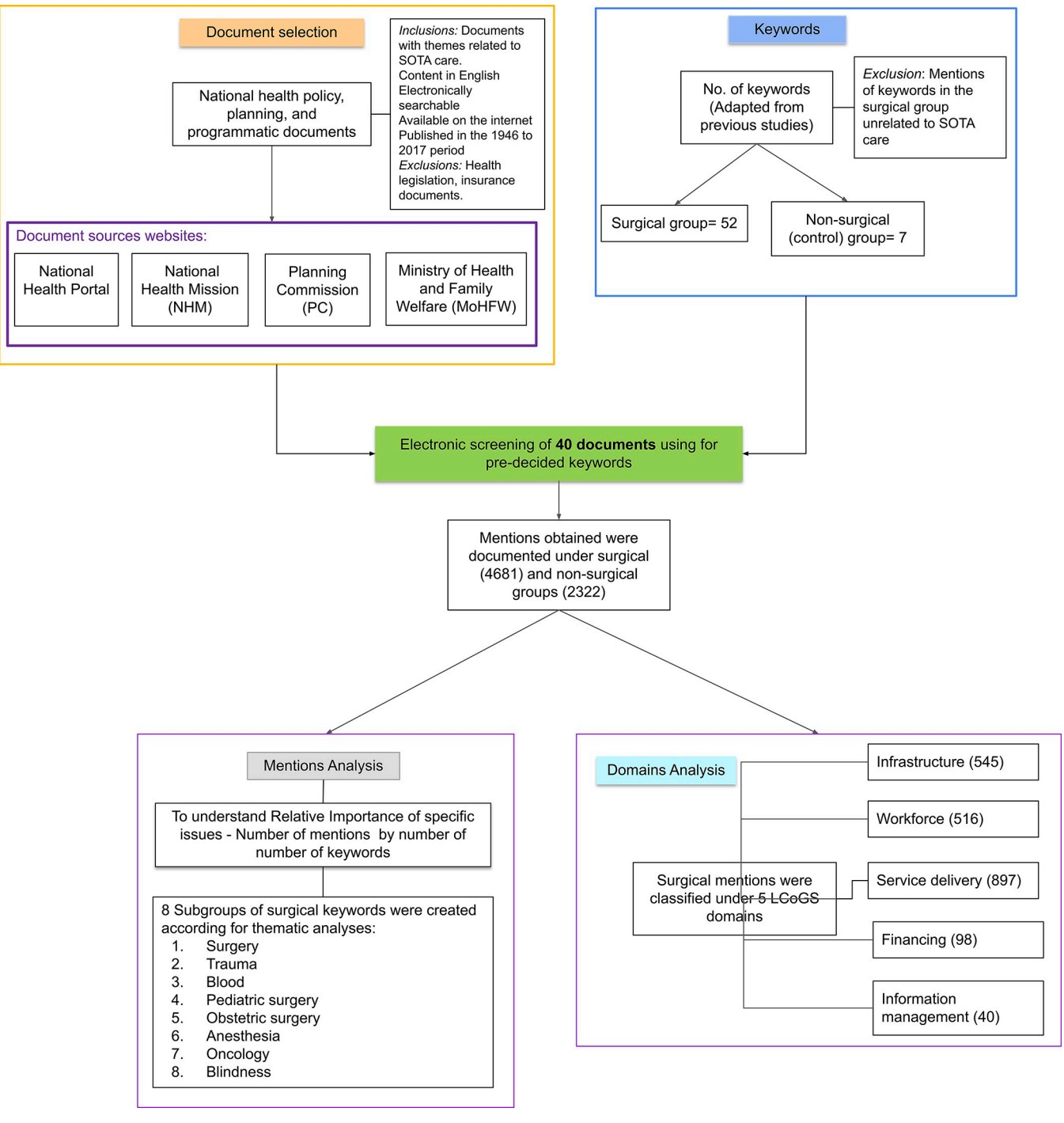

**Fig 1. Flow diagram depicting study analysis steps.**

obtain various committee reports which played an instrumental role in health policy planning in the early post-independence era. Next, the Planning Commission (PC) website was accessed to acquire the PC reports health sections (https://www.india.gov.in/website-planning-commission). The three National Health Policies were acquired from the National Health Portal (http://www.nhp.gov.in/). National Rural and Urban Health Mission related program

documents along with IPHS guidelines were accessed from the NHM website (https://nhm.gov.in/). Finally, using a snowballing approach, websites of various vertical programs that were relevant to SOTA care were accessed for programmatic guidelines or policy documents. A complete list of forty documents included in the current analysis is presented in **S1 Table**.

## 2.3 Keywords and document screening

The included documents were electronically searched for keywords and the relevant number of mentions obtained against each keyword search were recorded. Two groups of keywords: 52 surgical and 6 non-surgical keywords (**Table 1**) were adapted from previous studies investigating the prioritization of surgical care in the African continent [6] and the inclusion of pediatric surgery in national health plans globally [8]. For surgical keywords, eight subgroups were made by grouping keywords under the following themes: surgery, trauma, blood, pediatric surgery, obstetric surgery, anesthesia, oncology, and blindness. For non-surgical keywords, we chose communicable diseases of public health concern in India in alignment with previous studies [5, 8]. For surgical keywords, we excluded non-relevant mentions. Non-relevance was defined if a mention was vague and unrelated to the research topic (i.e. surgery-related terms or included public health concerns). For example, the keyword "transport" can arise in various contexts. However, we only included those mentions of "transport" which were related to the transport of surgical equipment or referral transport for surgical patients. Index and glossary terms were excluded from the final results for both the surgical and non-surgical keywords.

The surgical mentions were further classified into five domains (infrastructure, workforce, service delivery, financing, and information management) based on the LCoGS report and previous literature [1, 5]. We retrieved the sentences including the keywords as obtained through the electronic search of surgical keywords. At times, accompanying sentences were also retrieved to retain the context in which the keywords were used. These were then assessed and categorized into one or more suitable/closely relevant domains. Detailed descriptions of surgical domains are presented elsewhere [1], but a brief description is provided in **Table 2**.

We followed a two-step process: screening and validation. First, the initial screening for mentions was conducted by one investigator. Next, 5% (n = 126) of all the mentions were re-screened by a second investigator for validation. Screening for domain classification was performed by two investigators. A third investigator rescreened a random sample of 5% (n = 61) statements containing surgical mentions for validating if mentions were appropriately classified across the domains defined by LCoGS.

The dataset constructed for this study is presented in **S1 Data**.

## 2.4 Outcome variables and analysis

We measured relative prioritization as the average number of mentions per keyword for each group: surgical vs. non-surgical (**Eq 1**). This allowed comparisons across groups and tracking changes in prioritization over the years. For domains, the relative importance was assessed by the raw (absolute) number and percentages of surgical keyword mentions (**Eq 2**). Descriptive statistics were used to analyze prioritization.

$$Relative\ importance\ of\ a\ Group = \frac{Number\ of\ mentions\ of\ all\ keywords\ in\ the\ group}{Number\ of\ keywords\ in\ the\ group} * 100 \quad (Eq1)$$

$$Relative\ importance\ of\ a\ Domain = \frac{Number\ of\ mentions\ of\ surgical\ keywords\ for\ the\ domain}{Number\ of\ surgical\ keywords\ for\ all\ domains} * 100 \quad (Eq2)$$

**Table 1. List of keywords included in the analysis.**

| Sr. No | | Group | Subgroup |
|---|---|---|---|
| 1. | surg* | Surgical | Surgery |
| 2. | trauma | Surgical | Trauma |
| 3. | accident | Surgical | Trauma |
| 4. | road | Surgical | Trauma |
| 5. | transport | Surgical | - |
| 6. | RTA | Surgical | Trauma |
| 7. | fall | Surgical | Trauma |
| 8. | injur* | Surgical | Trauma |
| 9. | emergency | Surgical | Trauma |
| 10. | blood | Surgical | Blood |
| 11. | transfusion | Surgical | Blood |
| 12. | operat* | Surgical | - |
| 13. | OT | Surgical | - |
| 14. | OR | Surgical | - |
| 15. | operative delivery | Surgical | - |
| 16. | orth* | Surgical | - |
| 17. | open fracture | Surgical | Trauma |
| 18. | open fracture fixation | Surgical | Trauma |
| 19. | club foot | Surgical | Pediatric Surgery |
| 20. | amputation | Surgical | Surgery |
| 21. | wound | Surgical | Trauma |
| 22. | incis* | Surgical | Surgery |
| 23. | excis* | Surgical | Surgery |
| 24. | burn | Surgical | Trauma |
| 25. | cauter* | Surgical | - |
| 26. | obstetric | Surgical | Obstetric |
| 27. | C-section, cesarian, caesarean, cesarean | Surgical | Obstetric |
| 28. | MTP | Surgical | Obstetric |
| 29. | abort* | Surgical | Obstetric |
| 30. | EMOC | Surgical | Obstetric |
| 31. | EMONC | Surgical | Obstetric |
| 32. | sterilisation | Surgical | Obstetric |
| 33. | NSV | Surgical | Obstetric |
| 34. | *ectomy | Surgical | Surgery |
| 35. | *otomy | Surgical | Surgery |
| 36. | *stomy | Surgical | Surgery |
| 37. | curett* | Surgical | Obstetric |
| 38. | laparo* | Surgical | Surgery |
| 39. | anaesth*, anesth* | Surgical | Anesthesia |
| 40. | pediatric surg* | Surgical | Pediatric Surgery |
| 41. | hernia | Surgical | Surgery |
| 42. | inguinal hernia | Surgical | Surgery |
| 43. | circumcision | Surgical | Surgery |
| 44. | append* | Surgical | - |
| 45. | cancer | Surgical | Oncology |
| 46. | neoplasm | Surgical | Oncology |
| 47. | tumor | Surgical | Oncology |

(*Continued*)

**Table 1.** (Continued)

| Sr. No | | Group | Subgroup |
|---|---|---|---|
| 48. | malignancy | Surgical | Oncology |
| 49. | chemo* | Surgical | Oncology |
| 50. | onco* | Surgical | Oncology |
| 51. | cataract | Surgical | Blindness |
| 52. | blindness | Surgical | Blindness |
| 53. | TB & tuberculosis | Non-surgical | - |
| 54. | HIV | Non-surgical | - |
| 55. | immune deficiency | Non-surgical | - |
| 56. | AIDS | Non-surgical | - |
| 57. | malaria | Non-surgical | - |
| 58. | immunization | Non-surgical | - |

# 3 Results

## 3.1 Analysis of mentions

The total number of surgical mentions across 40 documents was 4681 compared to 2322 non-surgical mentions. Values of mentions per keyword for the surgical group were 90.02 and 387 for the non-surgical group (ratio = 1:3.5). This points to lower relative prioritization of surgical keywords. The root keyword 'surg*' had the maximum number of mentions (841) (**Fig 2A**) among the surgical group while 'malaria' had the most number of mentions (885) in the non-surgical group (**Fig 2B**). 'Blood' (459) and 'operat*' (333) had the most mentions following 'surg' while 'fall' and 'clubfoot' had just one mention across all documents (**Fig 2B**). Other frequently mentioned surgical keywords included 'emergency' (329) and 'obstetric' (221) (**Table 3**). Five (9.6%) surgical keywords (OR, operative delivery, open fracture, open fracture fixation, and EMONC) had no mentions across all documents.

The surgical keywords were categorized under the eight sub-groups. Among these, surgery (1160) had the most mentions while pediatric surgery (7) had the least number of mentions. Surgical prioritization showed a decline when assessed across the time period through which our data sources span (1946–2017) (**Fig 3A**), even when considering the documents published over the years (**Fig 3B**). This is despite minor variations in the number of documents published across the given time span (**Fig 3B**). The surgical and non-surgical groups seemed to follow similar trends across time with analogous peaks and drops across years. The highest overall surgical prioritization was in 2012 (38.19 mentions per keyword) and was lowest in the year 2014 (0.02 mentions per keyword).

**Table 2. Description of the LCoGS surgical domains [1].**

| Sr. No | Surgical domains as defined by LCoGS | Description |
|---|---|---|
| 1. | Infrastructure | Distribution of and access to surgical facilities, readiness for safe care provision, blood banking, and strengthening of referral systems. |
| 2. | Workforce | Training and recruitment of surgical physicians and nursing staff. |
| 3. | Service Delivery | Conveyance of surgical services, particularly the Bellwether procedures (i.e. laparotomy, open fracture fixation, and cesarean section), through a well-connected health system. |
| 4. | Financing | Inclusion of surgical packages under universal healthcare coverage and the apportioning GDP to surgical expenditures. |
| 5. | Information Management | Presence of intelligence systems for seamless monitoring of SOTA care. |

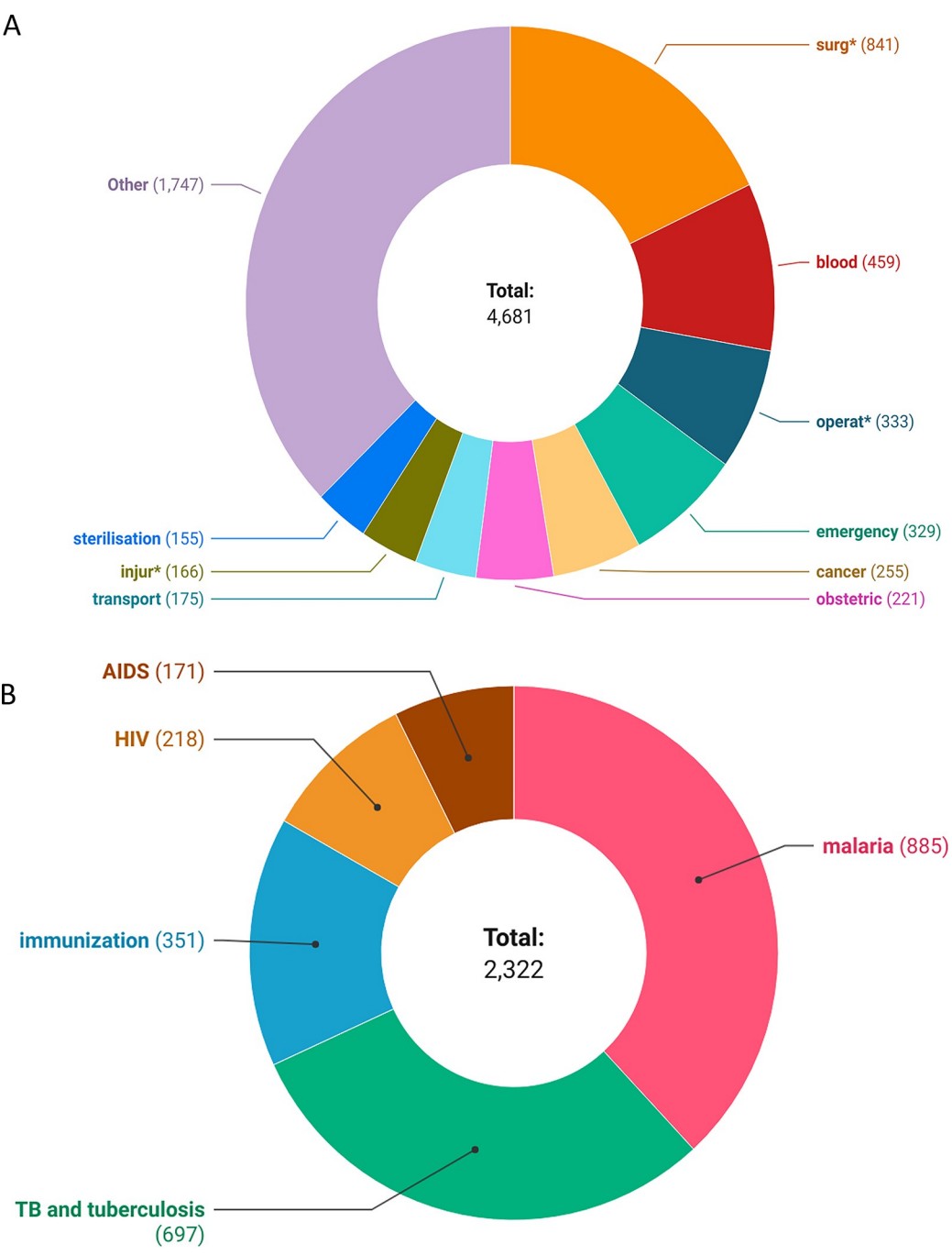

**Fig 2. A: Distribution of mentions by keywords included in the surgical group.** Ten keywords with the most mentions are displayed while other keywords are clubbed under 'Other'. Keywords with * refer to root keywords that can have multiple expanded forms. E.g., surg* can include surgical, surgery, surgeon, etc. **B: Distribution of mentions by keywords in the non-surgical group**.

Seventeen documents (42.5%) gave more weight to the surgical group when compared with the non-surgical (**Fig 4**). Seven of the 40 documents (17.5%) had no mention of the keyword 'surg*'. Two documents (5%) had no mention of any surgical keywords.

Surgical prioritization was greater in NHP 2017 [27] compared to the first (1983) [25] and second (2002) NHPs [26] with 3.18, 2.25 and 2.14 mentions per keyword, respectively

**Table 3. Absolute and relative mentions of each keyword included in the analysis.**

| Sr. No | Keyword | Subgroup | Number of absolute mentions | Relative importance of the keyword |
|---|---|---|---|---|
| 1. | surg* | Surgery | 841 | 16.17 |
| 2. | trauma | Trauma | 119 | 2.29 |
| 3. | accident | Trauma | 120 | 2.31 |
| 4. | road | Trauma | 62 | 1.19 |
| 5. | transport | - | 175 | 3.37 |
| 6. | RTA | Trauma | 3 | 0.058 |
| 7. | fall | Trauma | 1 | 0.019 |
| 8. | injur* | Trauma | 166 | 3.19 |
| 9. | emergency | Trauma | 329 | 6.33 |
| 10. | blood | Blood | 459 | 8.83 |
| 11. | transfusion | Blood | 133 | 2.56 |
| 12. | operat* | - | 333 | 6.40 |
| 13. | OT | - | 113 | 2.17 |
| 14. | OR | - | 0 | 0 |
| 15. | operative delivery | - | 0 | 0 |
| 16. | orth* | - | 80 | 1.54 |
| 17. | open fracture | Trauma | 0 | 0 |
| 18. | open fracture fixation | Trauma | 0 | 0 |
| 19. | club foot | Pediatric Surgery | 1 | 0.019 |
| 20. | amputation | Surgery | 9 | 0.17 |
| 21. | wound | Trauma | 43 | 0.83 |
| 22. | incis* | Surgery | 11 | 0.21 |
| 23. | excis* | Surgery | 43 | 0.83 |
| 24. | burn | Trauma | 66 | 1.27 |
| 25. | cauter* | - | 14 | 0.27 |
| 26. | obstetric | Obstetric | 221 | 4.25 |
| 27. | c-section,cesarian,caesarean,cesarean | Obstetric | 29 | 0.56 |
| 28. | MTP | Obstetric | 62 | 1.19 |
| 29. | abort* | Obstetric | 97 | 1.87 |
| 30. | EMOC | Obstetric | 8 | 0.15 |
| 31. | EMONC | Obstetric | 0 | 0 |
| 32. | sterilisation | Obstetric | 155 | 2.98 |
| 33. | NSV | Obstetric | 12 | 0.23 |
| 34. | *ectomy | Surgery | 136 | 2.62 |
| 35. | *otomy | Surgery | 68 | 1.31 |
| 36. | *stomy | Surgery | 35 | 0.67 |
| 37. | curett* | Obstetric | 3 | 0.06 |
| 38. | laparo* | Surgery | 35 | 0.67 |
| 39. | anaesth*, anesth* | Anesthesia | 114 | 2.19 |
| 40. | pediatric surg* | Pediatric Surgery | 6 | 0.12 |
| 41. | hernia | Surgery | 21 | 0.40 |
| 42. | inguinal hernia | Surgery | 7 | 0.13 |
| 43. | circumcision | Surgery | 8 | 0.15 |
| 44. | append* | - | 18 | 0.35 |
| 45. | cancer | Oncology | 255 | 4.90 |
| 46. | neoplasm | Oncology | 8 | 0.15 |
| 47. | tumor | Oncology | 16 | 0.30 |

(*Continued*)

**Table 3.** (Continued)

| Sr. No | Keyword | Subgroup | Number of absolute mentions | Relative importance of the keyword |
|--------|---------|----------|------------------------------|-------------------------------------|
| 48. | malignancy | Oncology | 8 | 0.15 |
| 49. | chemo* | Oncology | 28 | 0.54 |
| 50. | onco* | Oncology | 8 | 0.15 |
| 51. | cataract | Blindness | 60 | 1.15 |
| 52. | blindness | Blindness | 142 | 2.73 |
| 53. | TB & tuberculosis | - | 697 | 116.17 |
| 55. | HIV | - | 218 | 36.34 |
| 56. | immune deficiency | - | 0 | 0 |
| 57. | AIDS | - | 171 | 28.5 |
| 58. | malaria | - | 885 | 147.5 |
| 59. | immunization | - | 351 | 58.5 |

(**Fig 5A**). The difference between the mentions per keyword for the surgical and non-surgical groups was the highest in NHP 2002. NHP 1983 was the only document with more surgical than non-surgical mentions per keyword. However, most starkly, NHP 2017 had just one mention of 'surg*'.

Eleven of the 12 Planning Commission reports from 1951 to 2012 had more relative mentions of non-surgical keywords than surgical. There was a decline in relative prioritization with 4.5 mentions per keyword in the last report (PC 12) [44] compared to 7.8 in the first (PC 1) [45] (**Fig 5B**). Furthermore, the non-surgical group had more mentions per keyword than the surgical group for all PC reports except PC 9 [46].

The validation of mentions showed a 9.24% mismatch rate between the two investigators.

### 3.2 Analysis of surgical domains

Across the five LCoGS suggested domains for surgical capacity and access, the maximum number of mentions were for Service Delivery (897) followed by Infrastructure (545), Workforce (516), Financing (98), and Information Management (40) (**Fig 6**). The IPHS District Hospitals (2012) [34] document had the maximum number of mentions for the Infrastructure (141) and Service Delivery (398) domains among all documents while the Bhore Committee report (1946) [14–16] contributed the most mentions for the Workforce domain (258) (**Fig 7**). The Mukerji 1 report [18] had the maximum proportion of mentions for the Financing domain (19). Twenty-one (52.5%) documents had no domain-specific mentions for Financing while 27 (67.5%) had no mentions for Information Management. Analysis of domain mentions against the eight surgical subgroups showed maximum prioritization for the Surgery subgroup (**Fig 8**). Within Surgery, Service Delivery (400) had the maximum number of domain mentions. The Surgery subgroup had the maximum number of mentions for all domains, except Information Management for which the Obstetric subgroup (10) had the maximum number of mentions. The validation of domains showed a 39.34% mismatch rate between the two investigators.

## 4 Discussion

### 4.1 Summary of findings

This study attempts to quantify SOTA care prioritization through systematic document analysis. We found that SOTA care prioritization is relatively limited in the post-2010 documents when compared to the older ones that focus more on infrastructure and system-wide issues.

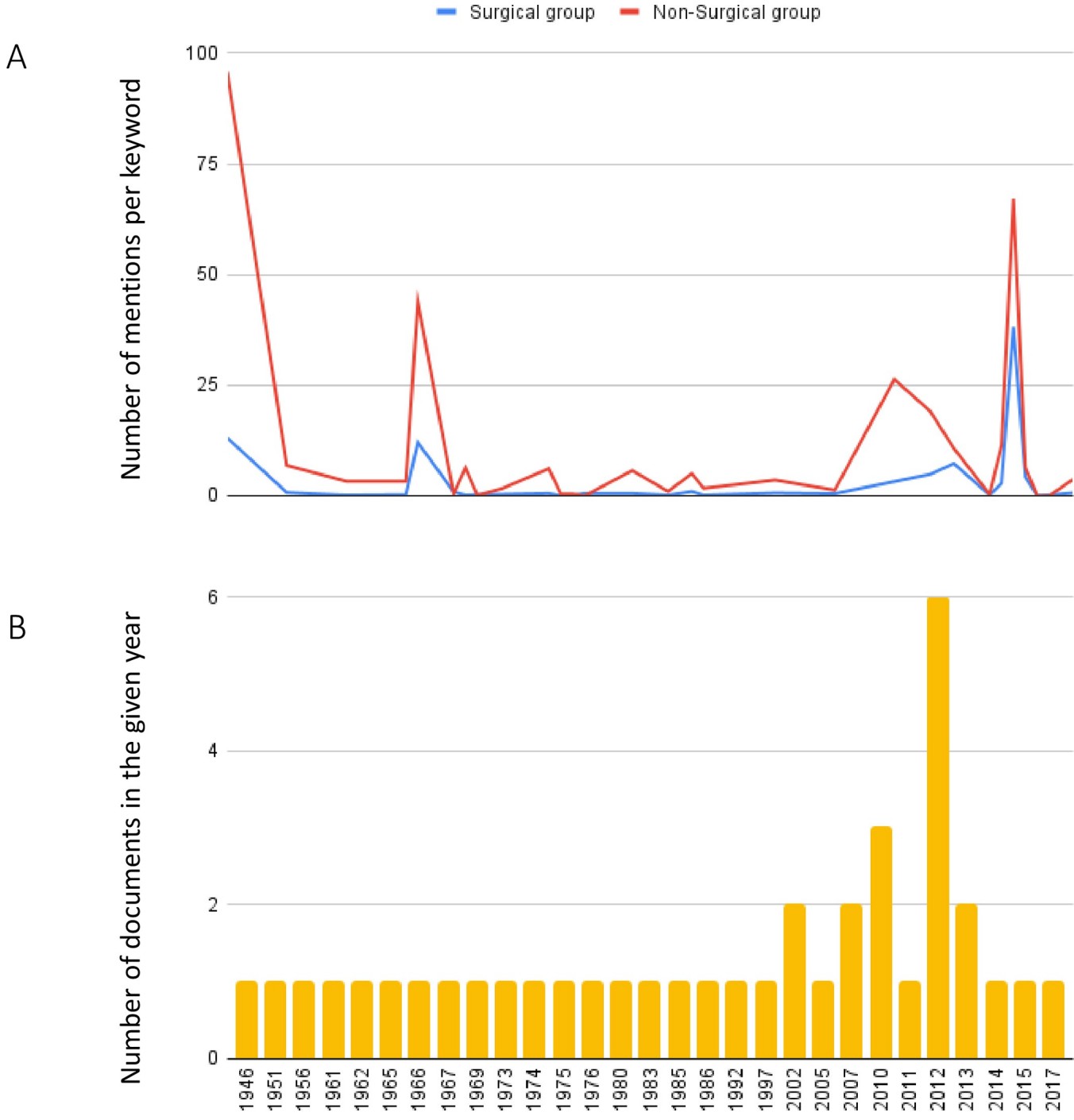

**Fig 3. A:** Surgical and non-surgical mentions per keyword from 1946 to 2017. **B:** Number of documents published in each year from 1946–2017.

Crucial policy documents such as the National Programme for Health Care of the Elderly (NPHCE) report and the National Council for Clinical Establishments (NCCE) report did not mention SOTA care at all while non-surgical issue mentions dominated other documents. Most importantly, the latest National Health Policy for India that was released in 2017 showed low SOTA care prioritization, and this policy is responsible for setting the country's current

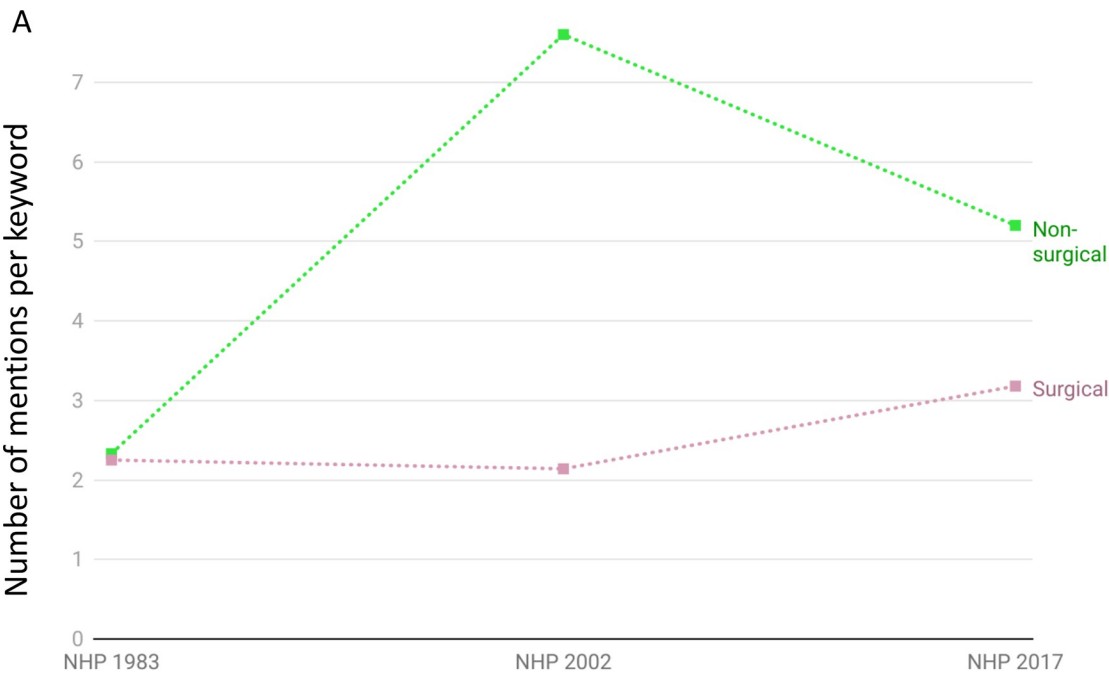

National Health Policy iterations from 1983-2017

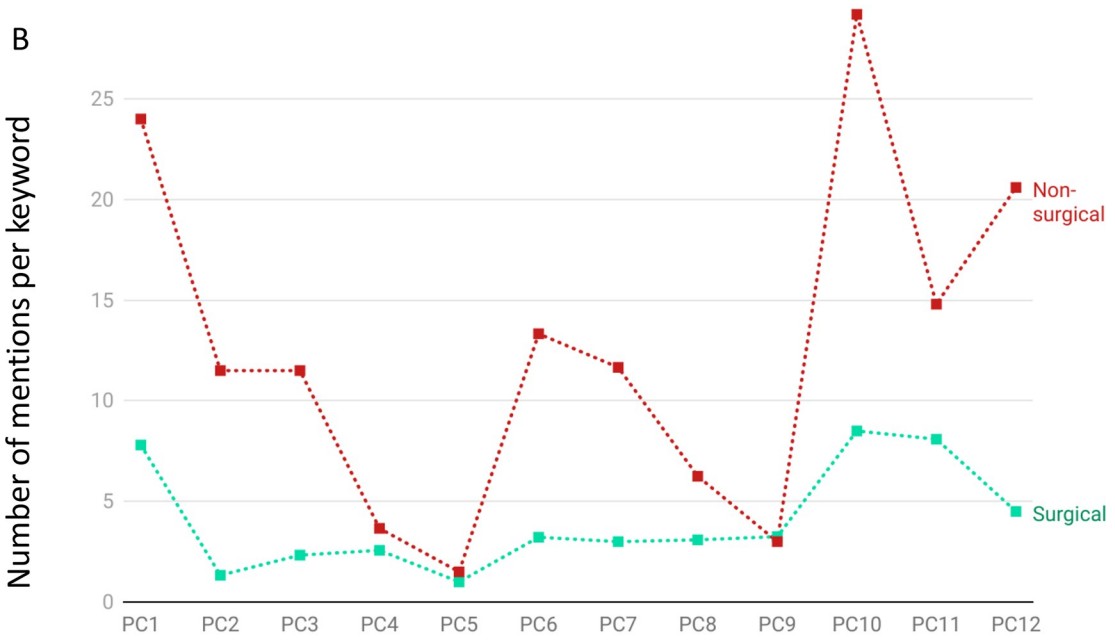

Planning Commission reports from 1951-2012

**Fig 4. Document-wise comparison of surgical and non-surgical mentions per keyword.** The documents are ordered in the decreasing order of the gap between surgical vs. non-surgical mentions per keyword. Abbreviations: PC = Planning Commission, IPHS = Indian Public Health Standards, NHP = National Health Policy, HLEG = High Level Expert Group, NRHM = National Rural Health Mission, NUHM = National Urban Health Mission, NBP = National Blood Policy, NPPMBI = National Programme for Prevention and Management of Burns and Injuries, NPCBVI = National Programme for Control of Blindness and Visual

Impairment, NPPC = National Programme for Palliative Care, NOTP = National Organ Transplant Program, NPHCE = National Programme for Health Care of the Elderly, NCCE = National Council for Clinical Establishments, NPCDCS = National Programme for Prevention and Control of Cancer, Diabetes, Cardiovascular Diseases, and Stroke.

agenda. Among the five LCoGS domains, information management was the least prioritized followed by financing. India has not mapped global surgery indicators, hence, investing in a robust information system is necessary.

## 4.2 Contextualizing with other literature

SOTA care prioritization using document analysis of National Health Strategic Plans was previously conducted for 43 countries in sub-Saharan Africa [6]. Nineteen percent of these plans had no SOTA care related mentions. This proportion was smaller in our study with 5% of documents with no mentions. The keyword 'Surg' had fewer mentions than 'cancer' in the analysis of the national health plans of the sub-Saharan African countries while the inverse is true for Indian policy documents. This is probably due to the inclusion of the programmatic and guidelines documents for NRHM, NUHM, and IPHS included in the current analysis that focuses predominantly on health infrastructure, health system components, and capacity building over specific disease conditions and population health targets.

Another analysis of national health plans from 124 countries evaluated pediatric surgery prioritization [8]. The findings of that analysis and the one presented here show an overall low number of mentions for keywords related to child surgery. Only 5% (2/40) of the documents showed mentions of the keyword 'pediatric surg' in our study, which was comparable to 7.3% (9/124) of documents found by Landrum and colleagues. No documents had more than 5 mentions of 'pediatric surg' in both cases depicting inadequate prioritization of pediatric surgery. Focus on the issue in policies can help initiate programmatic allocations and other steps downstream that can ultimately improve SOTA care delivery for pediatric populations.

We found that surgical workforce-related mentions accounted for about 24% of all mentions for different domains. Surgical workforce density is a global surgery indicator that can proxy a country's surgical care preparedness. Limited prioritization of the surgical workforce is in line with the importance given to the health workforce in Indian policies. An analysis of National Health Policy (NHP) documents previously noted that general health workforce-related recommendations have increased from 13 in 1983 to 37 in 2002 to 120 in 2017 [47]. In the current analysis, we found that surgical workforce mentions (not recommendations) across NHP documents increased from 0 in 1983 to 1 in 2002 to 3 in 2017. This depicts further lower priority given to the surgical workforce among limited attention to the overall health workforce.

## 4.3 Interpretation and relevance of findings

The current analysis depicts the need for greater prioritization of SOTA care issues in policy and programmatic documents across 70 years of health policymaking and planning in India. This is in stark contrast to the SOTA care disease burden and needs in the country. For instance, the population-level surgical rates range from as low as 341 to 3646 per 100,000 people, falling below the target of 5000 surgeries per 100,000 people recommended by the LCoGS [48–50]. While country-level data is absent, at the regional level in 2010, South Asia had a burden of over 29 million disability-adjusted life-years (DALYs) that could have been averted by access to essential and emergency SOTA care, which was close to the combined burden of tuberculosis (23 million DALYs), HIV (5 million DALYs), and malaria (7 million DALYs) [4,

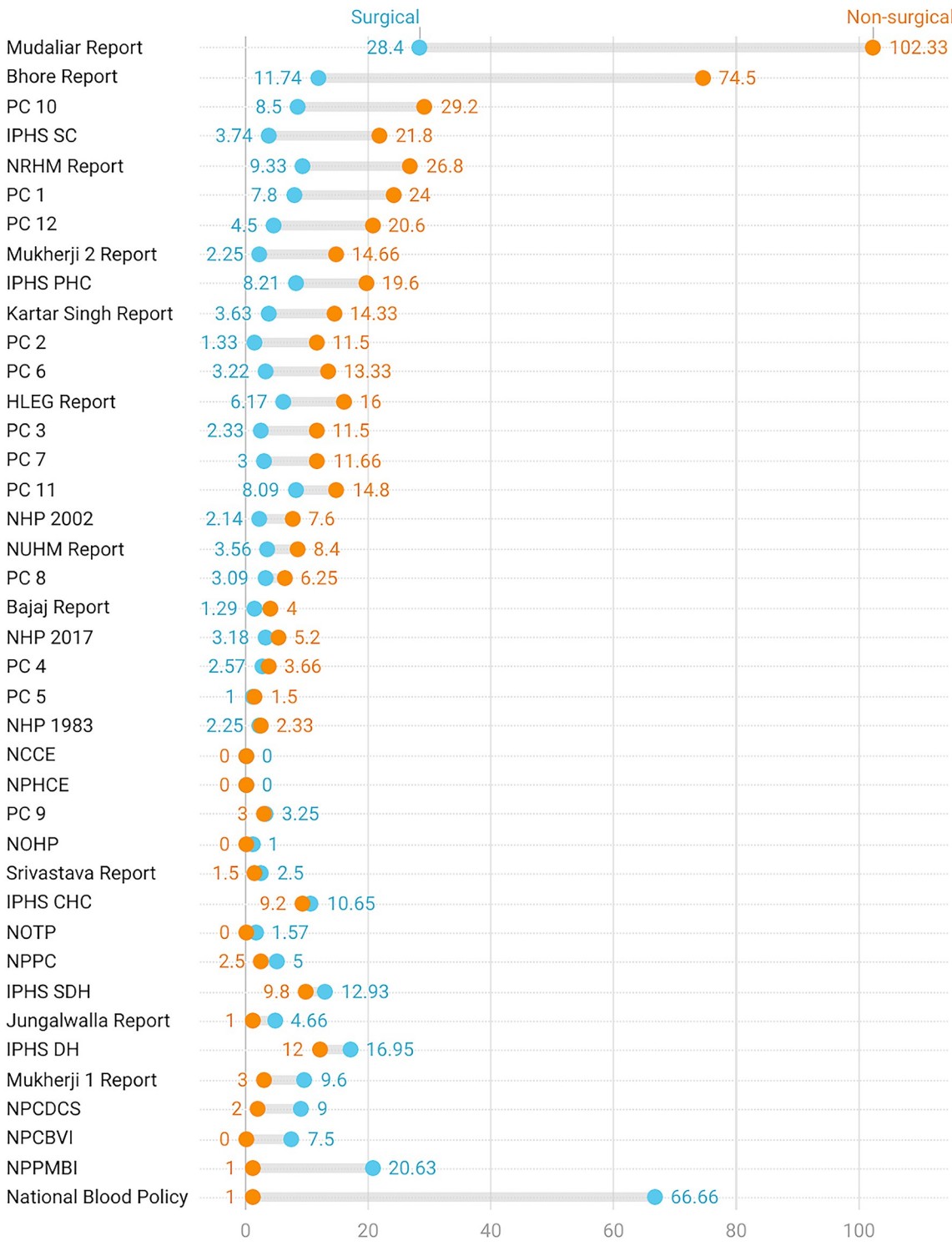

**Fig 5. Progression of relative prioritization assessed by mentions per keyword across A)** Iterations of the National Health Policy of India across 1983, 2002, and 2017 and **B)** Planning Commission (PC) reports.

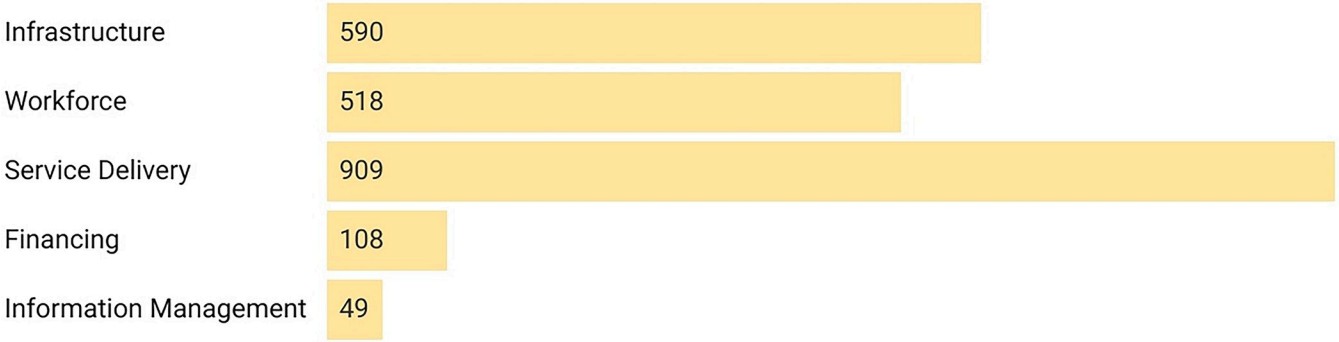

**Fig 6. Distribution of mentions of surgical keywords across the five surgical care domains defined by the Lancet Commission on Global Surgery (LCoGS).**

51]. Yet, the prioritization of SOTA care, recorded by the mentions per keyword, was only a fraction of that for communicable diseases.

We found that SOTA care was given more attention compared to non-surgical issues in Indian Public Health Standards (IPHS) guidelines for district hospitals, sub-district hospitals, and community health centers and standalone documents such as those for National Blood Policy, National Programme for Prevention and Management of Trauma and Burn Injuries (NPPMBI), and National Programme for Control of Blindness and Visual Impairment (NPCBVI) among others. While the latter two are clearly oriented towards SOTA care aspects, an important caveat in the case of National Blood Policy is that it focuses on blood supply and safety mainly in the context of the HIV pandemic among other health systems issues.

Attention across SOTA care issues was quite variable, with limited or no focus on some issues. For instance, there is almost no focus on pediatric surgery. This is starkly in contrast to the country's demographic characteristics and health needs. India has one of the world's largest pediatric populations and the highest under-five mortality rates. While there has been massive progress in under-five mortality reduction in the last three decades, India is still not on track to achieve the Sustainable Development Goal target for under-five mortality reduction by 2030. Investing in pediatric surgery can have society-wide economic benefits for the country [52]. The recent evidence also points to a significant reduction in neonatal and under-five mortality associated with the scale-up of the surgical workforce [53]. Second, there is no focus on the surgical workforce in National Health Policy documents. India faces a critical shortage of health workforce with only 26.5 doctors, nurses, and midwives per 10,000 population compared to the goal of 44.5 prescribed by the WHO [54]. Yet, the policy focus on the health workforce is limited and does not match the magnitude of deficits across different health workforce cadres [47]. Limited training spots largely concentrated in a few large urban centers make the surgical workforce scarce [55]. India requires 291,824 surgeons, obstetricians, and anesthetists by 2030 to meet the LCoGS target [56]. Such a scale-up would require dedicated policies, programs, and investments toward surgical training within the broader context of SOTA care planning. Third, while components of SOTA care are covered in IPHS guidelines and a few other programmatic documents, no mechanisms are described for financing SOTA care. Unlike several other LMICs, India lacks a comprehensive analysis of subnational estimates for out-of-pocket, catastrophic, and impoverishing expenses due to surgical care [57]. There are no clear mentions of budgetary allocations for SOTA care. Finally, there is very little focus on information systems for SOTA care across policy and programmatic documents, including the most recent National Health Policy of 2017. This might partly explain the lack of baseline assessment of LCoGS global surgery indicators in India [11].

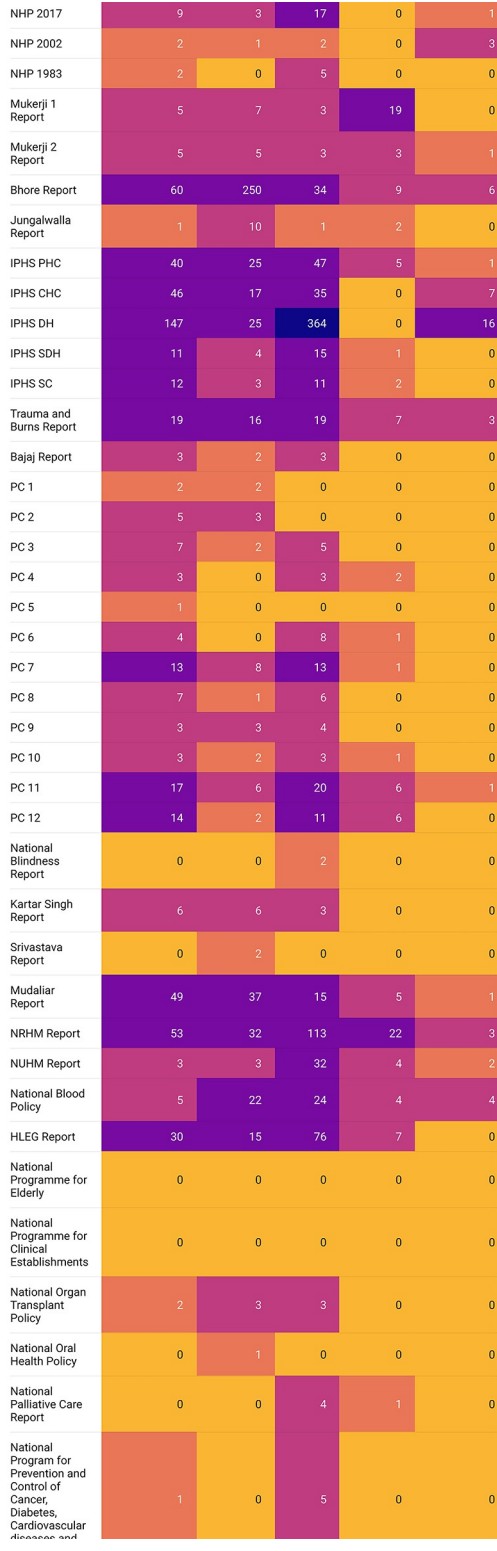

**Fig 7. Heatmap showing the document-wise distribution of the mentions under the five LCoGS-defined surgical care domains (infrastructure, workforce, service delivery, financing, and information management) across 40 documents.**

| Subgroup | Infrastructure | Workforce | Service Delivery | Financing | Information Management |
|---|---|---|---|---|---|
| Anesthesia | 2 | 13 | 1 | 0 | 0 |
| Blindness | 7 | 5 | 23 | 2 | 2 |
| Blood | 37 | 47 | 80 | 12 | 8 |
| Obstetrics | 27 | 41 | 53 | 7 | 10 |
| Oncology | 19 | 8 | 0 | 7 | 2 |
| Surgery | 145 | 272 | 400 | 25 | 6 |
| Trauma | 80 | 51 | 152 | 15 | 0 |
| Others | 228 | 79 | 157 | 30 | 4 |

**Fig 8. Heatmap showing the distribution of the LCoGS-defined domains across the eight subgroups of surgical keywords.**

Enhancing SOTA care prioritization in India's health policy agenda can be conducted through horizontal integration of SOTA care access issues across multiple policies and programs as has been prescribed in the NSOAP development process [57]. For instance, child health programs could include pediatric surgery coverage targets as it has been shown that scale-up of the pediatric surgical workforce is associated with reductions in child mortality [58]. Another way towards greater prioritization of SOTA care is through vertical policies and programs such as the National Programme for Prevention and Management of Trauma and Burn Injuries (NPPMBI). This program along with SOTA care relevant IPHS guidelines can pave the way for a comprehensive plan targeted towards the country's needs. Previously, several countries in Africa and South Asia have developed national surgical, obstetric, and anesthesia plans (NSOAPs) and national vision documents [7]. Additionally, evidence from Ethiopia suggests that NSOAP development and implementation can improve SOTA care service delivery [59]. Hence, developing NSOAP can help identify the needs unique to India, set targets for scale-up and appropriate distribution of resources, enable data collection for monitoring and evaluation, and create financing mechanisms for achieving the targets.

Multiple factors could underlie the limited prioritization of SOTA care. First, evidence on both the magnitude of the problem (e.g., disease and economic burden of surgically-treatable conditions, lack of access, scarcity of workforce, limited postoperative safety, etc.) and the potential solutions (e.g., the cost-effectiveness of emergency and essential procedures, technological innovations, societal benefits of SOTA care scale-up, etc.) has come up only recently prior to and during the Global Surgery movement [1]. Second, traditionally access to surgery has been projected and perceived as an elective luxury for a few instead of a basic component of emergency and essential services and a human right for all. Third, unlike several other issues such as tuberculosis, malaria, epidemics, primary healthcare, universal health coverage, etc. public health connections and implications of SOTA care have remained unexplored making its rise on the policy agenda difficult. Fourth, across countries, vertical policymaking, planning, and resource allocation end up focusing on specific diseases (or single issues) such as tuberculosis, HIV/AIDS, cancer, dementia, etc. This could lead to inadvertent neglect toward cross-

cutting areas such as SOTA care. Fifth, in India's case, infectious diseases such as tuberculosis, malaria, leprosy, smallpox, etc. have been a major public health concern and have occupied a larger chunk of the country's resources, leaving little room for SOTA care prioritization. Sixth, with the exception of the recent efforts [11], limited advocacy by major policy actors and entrepreneurs and missing political will for SOTA care could also explain low prioritization. However, beyond the speculations listed above, there is a need for a systematic assessment of reasons underlying low prioritization at policy development and implementation levels using existing frameworks (e.g., Shiffman and Smith framework, Kindon's multiple streams framework, etc.).

### 4.4 Strengths, limitations, and future directions

The current study has some notable strengths. First, a wide range of documents covering a span of seventy years including health committee recommendations, five-year plans, health policies, and programmatic guidelines. Second, the keywords used attempted to capture multiple aspects of SOTA care and were adapted from existing literature, making our findings comparable. The analysis for LCoGS-defined domains helps ascertain areas requiring immediate attention. Third, we counted mentions towards surgical only when those keywords were used in the context of SOTA care, thereby ensuring the rigor of analysis.

The current study also has several limitations. First, while an extended list of SOTA care related keywords was adapted from relevant studies, there is no clear categorization of the degree of relevance of each keyword to SOTA care. For instance, whether "blood" or "cancer" would be more important to prioritize under SOTA care cannot be decided in the current analysis. Future studies would benefit from expanding the keyword list and validating it through consensus among diverse shareholders. Second, using mentions per keyword is only one out of several metrics to investigate the prioritization of issues in policy documents. Future studies should consider using machine-based text mining approaches such as term frequency-inverse document frequency (tf-idf) to understand the relative importance of a keyword in a collection of documents. Giving differential systematic weights to keywords and grouping them according to relevance could be another approach to further investigate the prioritization of important keywords. Comparison of policy prioritization against objective metrics such as changes in disease burden or LCoGS indicators would help in the validation of the mentions-based approach. Third, we conducted a thematic classification of mentions based on the LCoGS-specified domains to assess the quantitative distribution, thereby missing out on exploring any SOTA care-related themes that emerge in these documents. In the future, a qualitative analysis such as investigating word clouds can help understand the themes emerging in these documents and the importance given to different SOTA care issues. Fourth, we included only 7 keywords in the non-surgical group based on the previous studies. However, there might be other non-surgical issues that have been highlighted in the Indian health policy and programmatic documents. Also, there might be health issues specific to the Indian context that we may have missed. However, this does not qualitatively change our main finding that SOTA care has received limited prioritization. On the contrary, it implies that the comparison of surgical vs. non-surgical mentions per keyword might be skewed more towards the surgical group and yet we observe a limited focus on SOTA care. Fifth, state-level health policy and programmatic documents were not included in the current analysis. Different aspects of health policymaking are governed exclusively or in a shared manner by national and state governments in India. For instance, issues such as accreditation of healthcare professionals and medical drug/device licensing are under the national bodies while several public health and epidemic management issues come under state-level legislature [60]. However, there are

multiple examples of health policymaking and financing in India using a top-down model, i.e., national policies leading to state policies, hence, an analysis of national documents is more pertinent. It should also be noted that state-wise analysis was not conducted due to limited data access due to the absence of a centralized comprehensive library of state-level documents. Sixth, we only included policy documents that were available in English, leaving out other Indian languages. However, almost all Indian documents released by the government of India have their content presented in both Hindi and English. Hence, it is unlikely that we may have missed any policies released solely in Hindi or any other regional language. Seventh, we could only include the documents that were available online. However, this was done to ensure the reproducibility of the analysis by using publicly available data that are easily accessible. Further, all national policies and planning documents are expected to be archived by different government ministries across digital platforms. Finally, our results are document dependent which means that inadvertent exclusion of relevant documents may skew the results. To ensure that this issue is minimal, we attempted to include relevant documents by screening multiple government platforms across ministries along with program-specific webpages.

### 4.5 Conclusion

This study demonstrates that India suffers from limited prioritization of SOTA care issues in policy and programmatic reports over the last seven decades, akin to other LMICs. The limited focus in the recent, post-2010 documents is particularly alarming. Information management and financing were the least prioritized among the LCoGS-defined domains for SOTA care planning. To achieve universal healthcare coverage, SOTA care needs to be focused. India needs to better integrate SOTA care issues in existing policies and move towards developing a National Surgical, Obstetric, and Anesthesia Plan.

### Supporting information

**S1 Table. List of documents included in the analysis.**
(DOCX)

**S1 Data. Dataset constructed for mentions and domains analysis.**
(XLSX)

### Author Contributions

**Conceptualization:** Ritika Shetty, Siddhesh Zadey.

**Data curation:** Ritika Shetty, Siddhesh Zadey.

**Formal analysis:** Ritika Shetty, Siddhesh Zadey, Anushka Jindal, Himanshu Iyer.

**Methodology:** Ritika Shetty, Siddhesh Zadey, Anushka Jindal.

**Project administration:** Siddhesh Zadey.

**Resources:** Sweta Dubey.

**Supervision:** Siddhesh Zadey.

**Writing – original draft:** Ritika Shetty, Siddhesh Zadey, Himanshu Iyer.

**Writing – review & editing:** Siddhesh Zadey, Anushka Jindal, Sweta Dubey, Gnanaraj Jesudian, Emily R. Smith, Catherine A. Staton, Tamara N. Fitzgerald, Joao Ricardo Nickenig Vissoci.

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
