## [Decision Letter · Decision Letter 0]

26 Jun 2023

PGPH-D-23-01008

Prioritization of Surgical, Obstetric, Trauma, and Anesthesia Care in India over Seven Decades: A Systematic Analysis of Policy Documents

Dear authors,

Thank you for submitting your manuscript to PLOS Global Public Health. After careful consideration, we feel that it has merit but does not fully meet PLOS Global Public Health’s publication criteria as it currently stands. Therefore, we invite you to submit a revised version of the manuscript that addresses the points raised during the review process.

Please consider the peer reviews and resubmit.

Please submit your revised manuscript as soon as possible.

If you will need more time than this to complete your revisions, please reply to this message or contact the journal office at globalpubhealth@plos.org. Please include the following items when submitting your revised manuscript:

We look forward to receiving your revised manuscript.

Kind regards,

Andreas K Demetriades, MBBChir, MPhil, FRCSEd, FEBNS.

Academic Editor

Journal Requirements:

2. We have noticed that you have uploaded Supporting Information files, but you have not included a list of legends. Please add a full list of legends for your Supporting Information files after the references list. 

Additional Editor Comments (if provided):

Reviewers' comments:

Reviewer's Responses to Questions

**Comments to the Author**

1. Does this manuscript meet PLOS Global Public Health’s publication criteria? Is the manuscript technically sound, and do the data support the conclusions? The manuscript must describe methodologically and ethically rigorous research with conclusions that are appropriately drawn based on the data presented.

Reviewer #1: Yes

Reviewer #2: Yes

2. Has the statistical analysis been performed appropriately and rigorously?

Reviewer #1: Yes

Reviewer #2: Yes

3. Have the authors made all data underlying the findings in their manuscript fully available (please refer to the Data Availability Statement at the start of the manuscript PDF file)?

Reviewer #1: Yes

Reviewer #2: Yes

4. Is the manuscript presented in an intelligible fashion and written in standard English?

Reviewer #1: Yes

Reviewer #2: Yes

5. Review Comments to the Author

Reviewer #1: Very interesting article highlighting the significant need to include SOTA priorities in policy documents to enact change at a national level. Would be interesting to see if adding additional surgical terms or adding weight to more important terms changes the findings.

Reviewer #2: This study aims to assess the level of prioritization afforded to surgical, obstetric, trauma and anesthesia care in health policy, planning and programmatic documents in India. Overall, the study is a useful contribution to the literature on global surgery policy, particularly in Asia where global surgery policy research has been lacking. The aims for the study are clear and the methodology used in appropriate, building on previous methodology used in other studies in Africa. It is also a more comprehensive analysis that previous studies. I commend the authors for this.

Minor edits/suggestions

- I understand that Hindi and English are the official languages of India. In focusing only on policies in English language, is it possible that some policies may have been excluded from the analysis?

- Did the authors consider reaching out to the Ministry of Health for additional policy documents that they might have missed by searching online? It is often the case that not all policy documents are available online.

- Overall the discussions and limitations mentioned are comprehensive and useful. However, it might be helpful if the authors could hypothesize why SOTA prioritization in health policy document is low in India despite the high burden of conditions requiring surgical care. They do mention the little focus on information management but perhaps there is more to this. The discussion section could be strengthened even further if the authors would consider expanding on why prioritization is low. Have there been any SOTA advocacy efforts in the past or currently which may explain the results? Perhaps the authors could consider using a framework like the Shiffman and Smith frame work on political priority for global health initiatives (https://www.thelancet.com/journals/lancet/article/PIIS0140-6736(07)61579-7/fulltext) or other similar frameworks to enhance the discussion of key findings.

- I commend the authors on a well articulated and comprehensive limititation section

6. PLOS authors have the option to publish the peer review history of their article (what does this mean?). If published, this will include your full peer review and any attached files.

**Do you want your identity to be public for this peer review?** For information about this choice, including consent withdrawal, please see our Privacy Policy.

Reviewer #1: No

Reviewer #2: No

---

## [Decision Letter · Decision Letter 1]

12 Jul 2023

Prioritization of Surgical, Obstetric, Trauma, and Anesthesia Care in India over Seven Decades: A Systematic Analysis of Policy Documents

PGPH-D-23-01008R1

Dear authors

We are pleased to inform you that your manuscript 'Prioritization of Surgical, Obstetric, Trauma, and Anesthesia Care in India over Seven Decades: A Systematic Analysis of Policy Documents' has been provisionally accepted for publication in PLOS Global Public Health.

Best regards,

Andreas K Demetriades, MBBChir, MPhil, FRCSEd, FEBNS.

Academic Editor

Congratulations

The peer review process recommends acceptance to publication

Reviewer Comments (if any, and for reference):

Reviewer's Responses to Questions

**Comments to the Author**

1. If the authors have adequately addressed your comments raised in a previous round of review and you feel that this manuscript is now acceptable for publication, you may indicate that here to bypass the “Comments to the Author” section, enter your conflict of interest statement in the “Confidential to Editor” section, and submit your "Accept" recommendation.

Reviewer #2: All comments have been addressed

2. Does this manuscript meet PLOS Global Public Health’s publication criteria? Is the manuscript technically sound, and do the data support the conclusions? The manuscript must describe methodologically and ethically rigorous research with conclusions that are appropriately drawn based on the data presented.

Reviewer #2: Yes

3. Has the statistical analysis been performed appropriately and rigorously?

Reviewer #2: Yes

4. Have the authors made all data underlying the findings in their manuscript fully available (please refer to the Data Availability Statement at the start of the manuscript PDF file)?

Reviewer #2: Yes

5. Is the manuscript presented in an intelligible fashion and written in standard English?

Reviewer #2: Yes

6. Review Comments to the Author

Reviewer #2: (No Response)

7. PLOS authors have the option to publish the peer review history of their article (what does this mean?). If published, this will include your full peer review and any attached files.

**Do you want your identity to be public for this peer review?** For information about this choice, including consent withdrawal, please see our Privacy Policy.

Reviewer #2: No
